# The Mediating Role of Physical Activity and Physical Function in the Association between Body Mass Index and Health-Related Quality of Life: A Population-Based Study with Older Adults

**DOI:** 10.3390/ijerph192113718

**Published:** 2022-10-22

**Authors:** Marcelo de Maio Nascimento, Élvio Rúbio Gouveia, Bruna R. Gouveia, Adilson Marques, Pedro Campos, Jesús García-Mayor, Krzysztof Przednowek, Andreas Ihle

**Affiliations:** 1Department of Physical Education, Federal University of Vale do São Francisco, Petrolina 56304-917, PE, Brazil; 2Department of Physical Education and Sport, University of Madeira, 9020-105 Funchal, Portugal; 3LARSYS—Laboratory for Robotics and Engineering System, Interactive Technologies Institute, 9020-105 Funchal, Portugal; 4Center for the Interdisciplinary Study of Gerontology and Vulnerability, University of Geneva, 1205 Geneva, Switzerland; 5Regional Directorate of Health, Secretary of Health of the Autonomous Region of Madeira, 9004-515 Funchal, Portugal; 6Saint Joseph of Cluny Higher School of Nursing, 9050-535 Funchal, Portugal; 7CIPER, Interdisciplinary Centre for the Study of Human Performance, Faculty of Human Kinetics, University of Lisbon, 1499-002 Cruz Quebrada, Portugal; 8Environmental Health Institute (ISAMB), University of Lisbon, 1649-020 Lisbon, Portugal; 9Department of Informatics Engineering and Interactive Media Design, University of Madeira, 9000-072 Funchal, Portugal; 10Public Health and Epidemiology Research Group, San Javier Campus, University of Murcia, 30720 San Javier, Spain; 11Institute of Physical Culture Sciences, Medical College, University of Rzeszów, 35-959 Rzeszów, Poland; 12Department of Psychology, University of Geneva, 1205 Geneva, Switzerland; 13Swiss National Centre of Competence in Research LIVES—Overcoming Vulnerability: Life Course Perspectives, 1015 Lausanne, Switzerland

**Keywords:** obesity, body mass index, quality of life, physical activity, physical function, aging, vulnerability

## Abstract

This study aimed (1) to investigate the association between body mass index (BMI), physical activity (PA), and physical function (PF) with health-related quality of life (HRQoL), and (2) to examine in-depth whether PA and PF mediate the relationship between BMI and HRQoL in older adults. We investigated 802 individuals (mean age 69.8 ± 5.6 years), residents of the Autonomous Region of Madeira, Portugal. Harmol and PA were assessed using the SF-36 and Baecke questionnaires, respectively, and PF by the Senior Fitness Test. The prevalence of overweight was 71.3%, 26.5% for normal weight, and 2.1% for underweight. We verified a small correlation between age and sex with BMI, PA, PF, and medium borderline with HRQoL. After adjusting for covariates (i.e., sex, age, comorbidities), the multivariate regression analysis indicated a 93.1% chance of improvement in HRQoL for low BMI, while PA and PF revealed a chance of increasing HRQoL by 91.8% and 60.0%, respectively. According to the serial mediation pathway, PA and PF partially mediated the association between BMI and HRQoL by 32.3% and 81.5%, respectively. The total variance of the model was 90%. It was concluded that BMI can negatively affect HRQoL. On the other hand, PA and PF are able to increase HRQoL levels during the aging process.

## 1. Introduction

The prevalence of obesity in the older adult population is a growing global public health problem worldwide [1,2]. Thus, the World Health Organization (WHO) has predicted for this century the “globesity” epidemic [3]. Obesity is a disease with multifactorial causes that seriously impact physical function and quality of life in the population of all age groups [4,5]. By 2030, a proportionate increase in the number of overweight and obese individuals is estimated for developing nations [6]. The main cause of this was attributed to the set of changes in sociodemographic characteristics and sedentary lifestyle habits [7]. For these reasons, there is great interest in understanding the negative mechanisms of obesity. Among the reasons, we highlight the recognition of obesity as a vulnerability risk factor capable of negatively affecting health-related quality of life (HRQoL) [8,9]. HRQoL is defined as the personal perception/sensation of well-being in mental, physical, and social domains of life [10].

Obesity is usually indicated by body mass index (BMI), and is associated with an increased risk of noncommunicable diseases [10,11]. Although BMI is a widely used relative weight measure, it has limitations because it does not consider important information, such as age, gender, bone structure, fat distribution or muscle mass [12]. It is noteworthy that older adults tend to undergo sequential changes in body composition, as well as a decline in stature, and these changes cannot be explained by BMI [13]. For this reason, when it comes to the estimation of nutritional status, it is suggested to associate with BMI measures of central adiposity, such as waist circumference (WC) and waist/hip ratio (WHR) [14].

Overweight and obesity are hot topics of interest in the health area, as their values allow us to better understand the mechanisms of different factors associated with the risk of developing a series of medical conditions [15], such as type 2 diabetes, linked to the metabolic syndrome [16], which in turn is associated with a constellation of symptoms, including hypertension [17] and coronary artery disease [18]. Moreover, obesity is also related to certain types of cancer [19]. Obesity is also capable of affecting the mobility of older adults, due to musculoskeletal problems [20], hindering their ability to remain independent and accident-free [21,22]. It is important to highlight that the association of vulnerable physiological aging with the aggravation of metabolic and genetic factors such as overweight and obesity facilitates the development of health problems [23,24]. Thus, obesity can not only increase the risk of morbidity, but also lead to premature death [22].

Over the years, sedentary behavior (SB) presents itself as one of the main factors responsible for overweight and obesity, determined by physical inactivity [25]. In contrast, adequate levels of physical activity (PA) can benefit the general physical and mental health state. For this reason, in 2020, the the World Health Organization (WHO) published Guidelines on PA and SB for all age groups [26]. This document recommended evidence-based health strategies, such as adequate levels of frequency, intensity, and duration of physical activity. In old age, maintaining PA levels is important because physiological aging, in addition to reducing metabolic functions, alters body composition and increases body weight [27]. It is known that daily PA levels tend to decrease [28], impairing physical function (PF) (i.e., aerobic resistance, muscle strength, flexibility, balance, gait speed) [29], consequently limiting older adults in performing their activities of daily living (ADLs) [30].

Low levels of PA and PF, as well as overweight and obesity can negatively affect the HRQoL of older adults [5,31]. The role that biological and psychological factors play in the etiology of quality of life were previously reported [9,32]. Thus, it is known that HRQoL is strongly related to perceived health, including symptoms of disease [33,34]. An admittedly effective measure to promote HRQoL consists of reducing body weight by the association between diet and increasing daily PA [35] levels, in addition to the weekly practice of physical exercise [36,37]. Proper levels of PA and PF are positively associated with a series of HRQoL aspects [38]. On the other hand, a high BMI value was negatively associated with HRQoL [39,40]. Studies with large populations sought to understand the relationship between overweight, obesity, and HRQoL [41]. However, so far, no study has examined the mediating role of PA and PF in the relationship between IMC and HRQoL of the older adult population resident in the autonomous region of Madeira, Funchal, Portugal. Health information of individuals living in areas far from large urban centers or unique cultural characteristics are essential for health surveillance services, as well as disease analysis and management [42].

Thus, we aimed (1) to investigate the association between BMI, PA, and PF with HRQoL, and (2) to examine in-depth whether PA and PF mediate the relationship between BMI and HRQoL in community-dwelling older adults. Our hypotheses were that (1) the higher the value of the BMI, the lower the participants’ HRQoL scores, and high levels of PA and PF will be able to enhance HRQoL scores; and (2) PA and PF variables will be able to partially mediate the relationship between BMI and HRQoL

## 2. Materials and Methods

### 2.1. Study Project and Participants

This was a cross-sectional analytical observational study. The analysis included 802 people, divided into 401 men and 401 women (69.8 ± 5.6 years). All participants resided in different districts of the Autonomous Region of Madeira, Funchal, Portugal. Recruitment was carried out between January and August 2007 through direct contact with the older adult population in public places (i.e., street markets, municipal gardens and churches), cultural and sports associations, clubs, day care centers, and nursing homes. Moreover, the study was also broadcast on local television, newspapers, and radio stations. The following inclusion criteria were considered: age between 60 and 85 years, living in the community, and being able to walk independently. The following exclusion criteria were also adopted: (1) some limitation in understanding or following the evaluation protocol, (2) pre-existing diseases such as cancer, Alzheimer’s, dementia, Parkinson’s, and (3) medical contraindications for submaximal physical exercises [43]. The study was approved by the Scientific Committee of the Department of Physical Education and Sport of the University of Madeira (UMa), and by the Regional Secretariat of Social Affairs Committees. Before starting the evaluations, all participants were informed about the procedures. Then, all participants received a consent form to sign. This study adhered to the Declaration of Helsinki. The set of assessments was carried out at the Laboratory of Human Physical Growth and Motor Development at UMa by a team of researchers previously trained to apply the assessment protocols.

### 2.2. Data Collection

#### 2.2.1. Demographics and Health Profile

Information regarding the number of falls in the last 12 months, daily medication consumption, sociodemographic data (gender, age), and comorbidities (self-reported history) were obtained through face-to-face interviews.

#### 2.2.2. Anthropometry

Using an anthropometric scale and a Welmy^®^ stadiometer calibrated to 0.1 cm and 0.1 kg [44], the participants’ body mass and height (H) were evaluated. BMI was defined as (weight [kilograms])/(height [m]^2^). BMI was classified into three categories, according to the cutoff points proposed by Lipschitz [45]: (i) low weight for BMI < 22.0 kg/m^2^; (ii) normal weight for BMI between 22.0–27.0 kg/m^2^, (iii) overweight for BMI > 27.0 kg/m^2^. A glass fiber metric tape with a 200 cm maximum length and a precision of 1 mm was used for waist circumference (WC) and hip circumference (HC) measurements. WC was measured in the minimal circumference of the abdomen in the zone between the inferior border of the last rib and the iliac crest. HC was measured by the greatest circumference between the iliac crest and the thighs, more specifically, at the level of maximum protrusion of the gluteal muscles. WC and HC measurements were taken twice and averaged. WC cutoff recommendations for overweight or obesity, and consequent association with disease risk, were calculated by the following cutoff points for men (WC ≥ 102 cm) and women (WC ≥ 88 cm) [46]. Moreover, waist-to-hip ratio (WHR) was estimated by WC (cm) divided by HC (cm). The interpretation of the WHR measurement was established using the sex-specific cutoff points recommended by the WHO [47], as follows: men (WC ≥ 0.90 m) and women (WC ≥ 0.85 m).

#### 2.2.3. Physical Activity

With the Baecke questionnaire [48], the PA of the participants was evaluated; for validation, see, for example, Ono et al. [49]. This instrument investigates participants’ PA levels based on the last 12 months, according to three specific domains: (1) work/domestic work (PA-work); (2) sports activities (PA-sport)—in this domain, only regular activities, carried out with a minimum duration of 1 h per week are evaluated; and (3) leisure time activities (PA-leisure). In the present study, the total score of the Baecke questionnaire (PA-total) was considered, calculated by the sum of (PA-work + PA-sport + PA-leisure)/3.

#### 2.2.4. Physical Function

PF was assessed with the Senior Fitness Test (SFT) [50]. The test battery includes five components (muscle strength, aerobic endurance, flexibility, agility/dynamic balance, and body-mass index (BMI)) and seven tests (chair stand (CST), arm curl (ACT), 6 min walk (MWT6), 2 min step test (2-mST), chair sit-and-reach (CSAR), back scratch (BST), and 8 ft up-and-go (FUG)). Other details on specificities of the evaluation procedures, namely, equipment, procedures, scoring, and safety precautions, can be found in the SFT manual. For the analyses, initially, the results of each of the seven tests were standardized by z-scores to obtain the same weight (total PF = CST + ACT + CSAR + BST + FUG + MWT6 + 2-mST). Afterwards, we calculated the sum of the scores of the seven indicators provided by the SFT, and used the PF of the study participants as a general indicator [51].

#### 2.2.5. Health-Related Quality of Life

To access the HRQoL, the Portuguese version [52] of the 36-item Short-Form Health Survey (SF-36 [53]) was used. This questionnaire is composed of a total of eight dimensions, categorized into physical and mental components, arranged as follows: (1) Physical: physical functioning (PF), physical role (PR), bodily pain (BP), and general state of health (GH); (2) Mental: vitality (VT), social functioning (SF), emotional role (ER), and mental health (MH). The SF-36 has scores that range from 0 to 100. Therefore, for interpretation, scores close to zero were assumed as a low perception of quality of life, while scores close to 100 represented a high perception of quality of life. In the present study, we used the SF-36 total score for the analyses, which was computed by the sum of the eight components of the questionnaire.

#### 2.2.6. Covariates

In this study, age, sex, hypertension, and diabetes were considered potentially confounding factors, and thus controlled for in the main analyses.

Although these variables were not of interest to the investigation, they may be directly associated with BMI (i.e., overweight and obesity); therefore, they could affect our response variables (PA, PF, HRQoL).

### 2.3. Statistical Analysis

Data distribution was verified using the Kolmogorov–Smirnov test. Thus, continuous variables were reported by means of mean and standard deviation (SD), and categorical variables by frequencies and percentages (%). The main characteristics of the participants were compared by three groups according to the IMC classification [45]. Statistical differences were determined by the chi-square test (categorical variables), and ANOVA (metric variables). Pairwise comparisons were performed by the Mann–Whitney U test, with Bonferroni corrections. First, we tested the strength and direction of the association between anthropometric variables (BMI, WC, WHR) with self-reported variables (gender, age, PA, HRQoL) and the objectively collected variable (PF). The analyses were performed by the Pearson correlation test, with interpretation by the following correlation coefficients.

In a second step, to verify effects, we performed a multivariate analysis. Thus, we estimate the associations between BMI, PA and PF (independent variables) with HRQoL (independent variable). All independent variables were included simultaneously in the regression model (enter method). Two models were tested: Model 1 unadjusted, and Model 2 adjusted for confounding factors (i.e., sex, age, comorbidities). In this analysis, residue independence and multicollinearity were verified by the Durbin Watson and Variance Inflation Factor statistics, respectively. Finally, a serial mediation analysis was performed to investigate whether PA and PF mediated the relationship between BMI and HRQoL (see Figure 1 for a general illustration). In recent decades, mediation analysis has developed considerably [54], as it allows the assessment of effects (different pathways and mechanisms). Unlike the multiple regression analysis, the mediation method was performed to investigate how the effect of BMI on HRQoL occurred, and more specifically, how one variable affected the other when the mediation variables (PA and PF) were entered into the model. Thus, it was possible to verify whether the relationship occurred due to the influence of the mediators, or if there was a legitimate relationship between BMI and HRQoL. Moreover, analysis of mediation can be useful for creating and testing theoretical models, as well as helping researchers to reflect initial propositions [55].

For the analyses, we assumed an indirect mediator effect when the causal effect of BMI (independent variable) was able to predict HRQoL (dependent variable) mediated by PA and PF [56]. We considered a mediation complete if, after the concomitant inclusion of PA and PF, the association between BMI and HRQoL was non-significant (when the confidence interval included zero). On the other hand, we assumed as partial mediation if, after the simultaneous inclusion of PA and PF, the association between BMI and HRQoL became weaker. The estimates of the mediation hypotheses were determined by the confidence interval (95%), with bias correction and acceleration (BCa) by the bootstrapping method with bias correction (5000 resamplings). All analyses were performed using the SPSS program. Mediation analyses were processed by PROCESS v4.0 [57], which is an add-on to the SPSS program. The results of the multiple regression analysis were presented by values of the odds ratio (OR), while the results illustrated in Figure 1 (mediation) corresponded to the standardized parameter *β*. In all analyses, a significance level of *α* < 0.05 was adopted.

## 3. Results

### 3.1. Main Characteristics of Participants

The characteristics of the 802 participants are presented in Table 1. The average age of both sexes showed that participants were in their sixties (women, 69.75 ± 5.64 years, and men, 69.87 ± 5.55 years) (*p* > 0.050). Regarding BMI classifications, the highest prevalence verified was for overweight (71.3%), followed by normal weight (26.5%) and underweight (2.1%). The average daily consumption of types of medication was 3.57 ± 2.56 (*p* = 0.018). Among the set of evaluated comorbidities, the most prevalent were hypertension, diabetes, visual impairment, hearing and musculoarticular problems (*p* < 0.050). According to the self-reports, the most common comorbidities found were visual impairment (61%), hypertension (50.9%), and hearing problems (24.7%). Regarding BMI, there was a mean of 29.51 ± 4.34, with a high prevalence of overweight in 42.5% and obesity in 43.8%. The WC average was 97.18 ± 11.27, and the WHR was 0.47 ± 0.71. Regarding PA, the mean observed was 7.30 ± 1.23, the mean PF score was 581.68 ± 144.20, while for HRQoL it was 68.57 ± 17.96. Regarding anthropometric variables, with the exception of height, all others indicated different results (*p* < 0.001). In general, the three BMI groups pointed to nearby PA levels (*p* > 0.050). Comparatively, older adults classified with normal weight had higher PF levels (602.44 ± 149.72) than those with underweight (589.80 ± 181.41) and overweight (573.55 ± 140.20) (*p* < 0001). Similarly, higher levels of HRQoL were found for normal weight participants (70.91 ± 18.17), followed by those with underweight (71.96 ± 10.09) and overweight (67.60 ± 17.96) (*p* < 0.001).

### 3.2. Correlations between the Main Variables of Interest in the Study

Table 2 presents the results of the correlation matrix. The findings revealed a positive and large association between BMI and WC (*r* = 0.760; *p* < 0.001), and positive and small correlation between BMI and sex (*r* = 0.170; *p* < 0.001), negative and medium correlation between BMI and PA (*r* = −0.380; *p* < 0.001) and also PF (*r* = −0.320; *p* < 0.001) and HRQoL (*r* = −0.462; *p* < 0.001). WHR indicated a positive and large association with measures of central adiposity BMI (*r* = 0.760; *p* < 0.001) and WC (*r* = 0.850; *p* < 0.001). On the other hand, WHR showed a negative and small association with all other variables: sex (*r* = −0.105; *p* < 0.010), PA (*r* = 0.−0.098; *p* < 0.001), PF (*r* = −0.096; *p* < 0.001), and HRQoL (*r* = −0.082; *p* < 0.001). Sex showed a positive and small association with PA (*r* = 0.250; *p* < 0.001) and a negative and small association with WC (*r* = −0.274; *p* < 0.001) and also PF (*r* = −0.284; *p* < 0.001). Consequently, sex pointed to a negative and borderline medium association with HRQoL (*r* = −0.301; *p* < 0.001). Age indicated a negative and small association with BMI (*r* = −0.033; *p* < 0.001), WHR (*r* = −0.085; *p* < 0.001), PA (*r* = −0.148; *p* < 0.001), and HRQoL (*r* = −0.191; *p* < 0.001), in addition to a negative and not significant association with WC (*r* = −0.066; *p* = 0.063) and gender (*r* = −0.010; *p* = 0.772). On the other hand, age showed a medium and negative association with PF (*r* = −0.406; *p* < 0.001). In turn, PA showed a positive and large association with PF (*r* = 0.511; *p* < 0.001) and HRQoL (*r* = 0.536; *p* < 0.001). Finally, a positive and large association was verified between PF and HRQoL (*r* = 0.591; *p* < 0.001).

### 3.3. Associations between BMI and HRQoL

Table 3 presents the associations and odds ratios between BMI and HRQoL. Our analysis resulted in a statistically significant model [F(3746) = 83,866; *p* < 0.001; R^2^ = 284]. BMI was negatively associated with HRQoL (OR = −0. 091, *t* = −5.48, *p* < 0.001). Therefore, a low BMI value represented a chance of increasing HRQoL by up to 90.9%. In turn, PA (OR = 0.055, *t* = 1.61, *p* = 0.038) and PF (OR = 0.453, *t* = 12,980, *p* < 0.001) indicated a positive association with HRQoL. Thus, having high levels of PA and PF represented a chance of promoting HRQoL by up to 94.5% and 54.7%, respectively. After adjusting for confounding factors (i.e., sex, age, hypertension, diabetes), the model remained significant [F(5744) = 13,379; *p* < 0.001, R^2^ = 312]. Thus, the association between BMI and HRQoL remained negative, showing that older adults with high BMI scores were more likely to have a lower level of HRQoL (OR = −0.069, *t* = −2.154, *p* < 0.001). This finding attested to the protective role that a low BMI value has for HRQoL, with a chance of improvement of up to 93.1%. In turn, PA and PF were positively associated with HRQoL (OR = −0.082, *t* = 2.440, *p* = 0.015 and OR = 0.400, *t* = 10.556, *p* < 0.001, respectively). In proportional terms, PA and PF indicated the chance of increasing HRQoL by 91.8% and 60.0%, respectively.

### 3.4. Mediation Analysis

Figure 1 presents the results of the serial mediation analysis. The model obtained was significant, predicting the two variables F(4.745) = 5.8742, *p* < 0.001, R^2^ = 0.18. Model 1 was controlled for potential confounders (i.e., sex, age, height, waist and hip circumference) and showed that BMI (independent variable) had a negative and significant association with the PA mediator (*β* = −0.03, *t* (745) = −2.656, *p* = 0.008) and negative and significant association with the PF mediator (*β* = −4.95, *t* (744) = −4.875, *p* < 0.001). According to our analysis, the proportion of the total effect of BMI on HRQoL, when PA and PF mediators were inserted, was approximately 32.3% and 81.5%, respectively, while the total variance of the model was 90%. Therefore, the estimated degree of association between PA and PF mediators was positive and significant (*β* = 40.03, *t* (744) = 11.226, *p* < 0.001). Model 2 showed significant and positive associations between PA and HRQol (*β* = 1.21, *t* (743) = 2.481, *p* = 0.013), as well as a positive and significant association between PF and HRQoL (*β* = 2.24, *t* (743) = 10.420, *p* < 0.001). In terms of the direct effect estimated by the model (BMI-HRQoL), it showed a negative and significant relationship (*β* = −0.28, *t* (743) = −2.181, *p* < 0.029). Likewise, the total effect (BMI-HRQoL) revealed a negative and significant association (*β* = −0.61, *t* (745) = −4.344, *p* < 0.001). Our statistical procedures tested three effects: (1) indirect path BMI-PA-HRQoL, which showed a negative and significant association (*β* = −0.03, SE (0.0183), 95% CI BCa = −0.0745–0.0033); (2) indirect pathway BMI-PF-HRQoL with negative and significant association (*β* = −0.24, SE (0.0563), 95% CI BCa = −0.3486–0.1300); and (3) total indirect pathway between BMI and HRQoL through PA and PF, which indicated a negative and significant partial mediation effect (*β* = −0.05, SE (0.0220), 95% CI BCa = −0.0983–0.0128).

## 4. Discussion

Our study aimed to investigate the association between the variables BMI, PA, and HR with HRQoL, and to examine in depth whether PA and PF mediate the relationship between BMI and HRQoL in community-dwelling older adults. The findings showed that 71.3% of older adults were overweight, which indicates potential for the aggravation of noncommunicable diseases [10,11]. Therefore, 26.5% of the participants indicated normal weight, and only 2.1% underweight. Correlation analyses showed that anthropometric variables (BMI, WC, WHR) were positively and largely related to each other, and negative and small with PA, PF and HRQoL. On the other hand, WHR indicated a negative and medium correlation with HRQoL. An interesting finding of the present study was that the variables age and sex showed small, and in some cases, non-significant, levels of correlation with the other variables. These results were not in agreement with previous studies; in general, age and sex are considered to influence the levels of PA [58,59], PF [60,61], and body weight [14,62] and, in turn, affect HRQoL levels [39,40].

Our first hypothesis was confirmed by multivariate analysis. When controlled for confounding factors (i.e., sex, age, hypertension, diabetes), it was found that a low BMI value presented a chance of up to 93.1% for participants to have a high HRQoL. The finding is in agreement with previous studies that highlighted the negative effects that high weight has on the health perception of the older adult population [63,64]. Moreover, we also confirmed that high levels of PA and PF increased the chances of older adults to have a high HRQoL by approximately 91.8% and 60.0%, respectively. These results are in agreement with previous studies carried out with the older adult population [65,66]. Regarding the mediation analysis, when PA and PF were inserted into the equation, the model revealed a negative and significant relationship between BMI and HRQoL. Thus, PA and PF were able to partially mediate these relationships by approximately 32.3% and 81.5%, respectively. Moreover, the total variance in HRQoL explained by the entire model was 90%. To our knowledge, this is the first study that has estimated in detail the mediation exerted by PA and PF in the association between BMI and HRQoL in the older adult population of the Autonomous Region of Madeira, Funchal, Portugal.

The literature highlights that obese older adults tend to have lower HRQoL compared to those with normal weight [5,21]. Our findings are in agreement with previous studies that found negative and significant associations between BMI (overweight and obesity) and HRQoL in the older adult population [39,40]. One of them was a cohort type carried out in England in two waves [40]. The authors confirmed that there is a clear inverse relationship between weight gain and HRQoL decrease, highlighting that among obese individuals, the time variable has a significant mediating role in the reduction of HRQoL. Furthermore, a high BMI remained independently related to HRQoL, denoting that obese but healthy individuals may be in transition to vulnerability and a worse future health status. This supposed independent relationship between obesity and HRQoL is consistent with the “healthy obesity” hypothesis [67], which reflects a subset of the older adult population that is transitioning to unhealthy obesity. In this context, a possible additional underlying process of the transition from a healthy to an unhealthy state may occur through the metabolic syndrome [68].

In the older adult population, overweight and obesity are responsible for an increased risk of chronic diseases, including cardiovascular diseases, arthritis, diabetes, and lung diseases [10,69]. All of these diseases potentiate the development of physical disability [22], resulting from an intolerance to exercise, which can also be associated with frailty and social or psychological problems [70]. Furthermore, considering the age-related etiology of PF, changes in body composition, especially sarcopenia, also play a strong role [71,72]. In a cohort study, it was found that in over 7 years of follow-up, participants without disability (IADL scale) but with sarcopenic obesity were 2.5 times more likely to report a disability in their IADL than those without sarcopenic obesity [73].

Our mediation analysis showed that for each 0.1 kg/m^2^ increase in the standard deviation (SD) of the BMI, the PF was reduced by 4.95 times. The findings are in agreement with previous evidence on obese older adults residing in the community, which pointed to the existence of a positive association between PF and factors related to physical and mental health (HRQoL). Notably, low PF levels result in a marginal perception of quality of life [63,71]. Moreover, we found that for every 0.1 kg/m^2^ increase in the standard deviation (SD) of BMI, PA was reduced by 0.30 times. The finding confirms that overweight/obesity potentiates the individual’s intolerance to increased mobility and daily energy expenditure, which is also directly associated with intolerance to regularly practice physical exercises [71,74]. Therefore, it is important that older adults are encouraged to engage in daily tasks that require body movement [37,75]. Consequently, there will be an increase in caloric expenditure, benefiting their health status as a whole. Both the WHO [26], and systematic review studies [7,76] advocated that high/medium levels of PA are essential for the consolidation of healthy aging, with a lower risk of cardiovascular diseases, accompanied by physical and mental well-being.

Controlling for confounding factors (i.e., sex, age, height, comorbidities, waist and hip circumference), it was found that for every 0.1 kg/m^2^ increase in standard deviation (SD), HRQoL increased by 1.21 times. Regarding the PF, it was observed that for each increase of 0.1 kg/m^2^ in the standard deviation (SD), the HRQoL increased 2.24 times. These results can expand the understanding of how PA and PF act on the components of the HRQoL of the older adult population. Previous studies have highlighted the role of different vigorous and non-vigorous leisure activities in treating and preventing the worsening of obesity [36,77]. Thus, increasing PA levels, both in the practice of weekly physical exercises, as well as in leisure and domestic activities, is an effective strategy to control and/or reduce body weight and, consequently, reduce other risk factors associated with obesity [77].

An increase in PA levels can improve PF, presumably benefiting variables associated with mobility such as muscle strength, balance, and cardiovascular endurance [75,78]. In advanced age, the interdependence between PA and PF is considered a determining factor for the individual’s autonomy and, therefore, crucial for the safe performance of IADL [79]. Thus, different types of physical exercises have been suggested to promote PA, PF and HRQoL, and consequently combat overweight/obesity [7,22]. Multicomponent exercise, for example, was suggested due to the breadth of training, which comprises flexibility, balance, and aerobic and resistance exercise in a single session [24]. Regarding the volume and intensity of physical training, the WHO recommends for adults, including those aged >64 years [26]: (1) at least 150 to 300 min of moderate-intensity aerobic PA, (2) at least 75–150 min of vigorous-intensity aerobic PA, or (3) to substantially benefit general health (physical and mental) with an equivalent weekly combination of moderate and vigorous-intensity activity.

Among the strengths of this study, we can mention the use of a large cohort, the inclusion of anthropometric variables objectively measured through standardized protocols, as well as an assessment of HRQoL, obtained through a validated instrument. In addition, the mediation analysis was adjusted for potential confounders (i.e., sex, age, height, comorbidities, waist and hip circumference). However, our study has some limitations that should be highlighted. First, the cross-sectional design does not allow for causal conclusions. Thus, we cannot determine when the observed PA and PF levels were established in relation to the verified BMI. Moreover, a low level of PF can limit mobility, which in turn can lead the individual to reduce their PA levels, increasing body weight [80]. Second, it is known that BMI is not a measure capable of providing complete information about the distribution of body fat [13]. Therefore, depending on the population evaluated, other factors such as sociodemographic factors, living conditions, and access to health information must be considered. Third, issues such as diet, sleep quality and/or daytime sleepiness, which are associated with obesity and WC, were not investigated. Fourth, the present study did not include variables related to functional capacity, such as dynapenia, sarcopenia and frailty, which are associated with age-related health changes and problems caused by body fat. Thus, we suggest that future studies include these variables and that investigations adopt longitudinal follow-up. Possible topics for investigation would be to observe the mediating role of PA and PF in the relationship between BMI and HRQoL, including moderators such as sociodemographic factors (i.e., income, race, sex, education), dietary habits, types of physical exercise, as well as weekly training frequency.

## 5. Conclusions

Our results provide insights into the mechanisms of vulnerability in a large sample of older adults with unique cultural, biological and psychosocial characteristics, residing in the Autonomous Region of Madeira, Funchal, Portugal. Therefore, the analyses showed the important mediating role of PA and PF in the relationship between BMI and HRQoL. Overall, high body weight was found to be negatively related to HRQoL. Therefore, in the case of the older adult population, increasing daily caloric expenditure, either through physical exercise or by intensifying leisure and domestic activities, can be a strategy capable of controlling and/or reducing body weight. Moreover, considering that we are in a post-pandemic period (COVID-19), that older adults were the main risk group for this serious disease worldwide, and that obesity was a potentiating factor in the risk of death from the virus, the detailed information in this study may be useful for monitoring and treating obesity, reducing vulnerability and promoting HRQoL.

## Figures and Tables

**Figure 1 ijerph-19-13718-f001:**
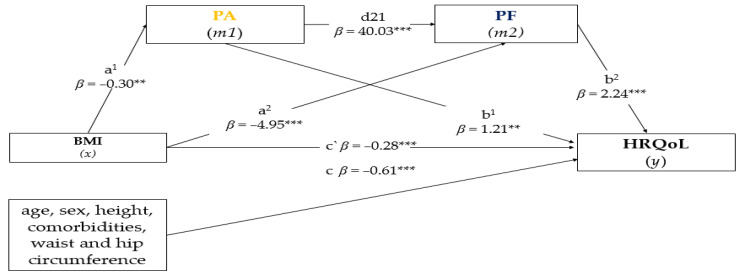
Analysis of parallel mediation of the effects of body mass index (BMI) on health-related quality of life (HRQoL) through physical activity (PA) and physical function (PF). The analysis was based on 5000 bootstrap samples. Results of the simultaneous regression analyses were presented by beta values (*β*), using the bootstrap method with bias correction for 5000 samples, followed by confidence intervals (95%). Path a^1^ and a^2^ = association of the BMI with PA and PF, respectively; Path d21 = association between PA and PF mediators; Path b^1^ and b^2^ = association of PA and PF mediators with HRQoL, respectively; Path c’ = direct effect (BMI—HRQoL), Path c = indirect effect: association of BMI with HRQoL through PA and PF mediators. ** *p* < 0.01, *** *p* < 0.001.

**Table 1 ijerph-19-13718-t001:** Main characteristics of the sample.

Variable	Full Sample (*n* = 802)	Underweight (*n* = 17)	Normal Weight (*n* = 213)	Overweight (*n* = 572)	*p*-Value
Age (years)	69.81 ± 5.60	71.64 ± 6.00	69.80 ± 5.90	69.76 ± 5.46	0.396
60–69	364 (45.4)	8 (47.1)	102 (47.9)	254 (44.4)	
70–79	408 (50.9)	8 (47.1)	101 (47.4)	299 (52.3)	
≥80	30 (3.7)	1 (5.9)	10 (4.7)	19 (3.3)	
Sex *n* (%)					0.011
Male	401 (100.0)	11 (64.7)	123 (57.7)	267 (46.7)^,^	
Female	401 (100.0)	6 (35.3)	90 (42.3)	305 (53.3)	
Falls (*n*)	1.74 ± 0.80	1.65 ± 0.86	1.71 ± 0.81	1.75 ± 0.80	0.735
Medication (*n*)	4.66 ± 0.98	5.35 ± 0.86	4.89 ± 0.94	4.55 ± 0.98	0.018 *
Comorbidities (%)					
Hypertension	408 (50.9)	13 (76.4) ^b,c^	136 (23.7) ^a,c^	259 (45.2) ^a,b^	0.038
Diabetes	188 (23.4)	6 (35.2) ^b,c^	72 (33.8) ^a,c^	110 (19.2) ^a,b^	0.028
Visual impairment	489 (61.0)	17 (100.0) ^b,c^	181 (85.0) ^a,c^	291 (50.9) ^a,b^	0.018
Hearing problems	198 (24.7)	6 (35.2) ^b,c^	91 (42.7) ^a,c^	101 (17.6) ^a,b^	0.024
Musculoarticular problems	46 (5.7)	9 (53.0) ^c^	12 (5.6) ^c^	25 (4.4) ^a,b^	0.044
Anthropometry					
Height (kg)	159.05 ± 8.69	159.14 ± 7.34	159.60 ± 8.16	158.84 ± 8.91	0.553 *
Weight (cm)	74.77 ± 13.06	50.85 ± 5.68 ^b,c^	63.88 ± 7.75 ^a,c^	79.54 ± 11.50 ^a,b^	<0.001 *
BMI (kg/m^2^)	29.51 ± 4.34	20.01 ± 1.02 ^b,c^	25.00 ± 1.48 ^a,c^	31.47 ± 3.39 ^a,b^	<0.001 *
WC (cm)	97.19 ± 11.27	76.30 ± 5.38 ^b,c^	87.67 ± 7.26 ^a,c^	101.35 ± 9.68 ^a,b^	<0.001 *
Men ≥ 102 cm (f)	197 (49.1)	—	6 (3.0)	191 (97.0) ^b^	<0.001 *
Women ≥ 88 cm (f)	358 (89.3)	—	22 (7.5)	272 (92.5) ^b^	<0.001 *
HC (cm)	101.47 ± 9.04	85.52 ± 3.92 ^b,c^	93.97 ± 4.61 ^a,c^	104.47 ± 8.20 ^a,b^	<0.001 *
WHR (cm)	0.95 ± 0.08	0.89 ± 0.69 ^b,c^	0.93 ± 07 ^a,c^	0.97 ± 0.95 ^a^	<0.001 *
Men ≥ 90 cm (f)	392 (97.8)	10 (2.6)	116 (29.6) ^a,c^	266 (67.9) ^a,b^	0.036 *
Women ≥ 85 cm (f)	—	—	—	—	—
PA (*n*)	7.30 ± 1.23	7.33 ± 1.12	7.42 ± 1.28	7.26 ± 1.21	0.122
PF (*n*)	581.58 ± 144.21	589.80 ± 181.41	602.44 ± 149.72 ^c^	573.55 ± 140.20 ^b^	0.031 *
HRQoL (*n*)	68.57 ± 17.96	71.96 ± 10.09	72.91 ± 18.17	67.60 ± 17.96	0.036 *

BMI: body mass index; WC: waist circumference; HC: hip circumference; WHR = waist-to-hip ratio; HRQoL: health-related quality of life; PA: physical activity; PF: physical function; Kg: kilogram; cm: centimeter; Kg/m^2^: kilogram divided by meter squared. * ANOVA *p* < 0.050; U Mann–Whitney ^a,b,c^ *p* < 0.050.

**Table 2 ijerph-19-13718-t002:** Correlation matrix coefficients between the studied variables.

Variable	1	2	3	4	5	6	7
1. BMI	1.00						
2. WC	0.760 **	1.00					
3. WHR	0.930 **	0.850 **	1				
4. Sex	0.170 **	−0.274 **	−0.105 *	1.00			
5. Age	−0.033 ^ns^	−0.066 ^ns^	−0.085 *	−0.010 ^ns^	1		
6. PA	−0.380 **	−0.288 **	−0.098 **	0.250 **	−0.148 **	1.00	
7. PF	−0.320 **	−0.254 **	−0.096 **	−0.284 **	−0.406 **	0.511 **	1
8. HRQoL	−0.462 **	−0.321 **	−0.082 **	−0.301 **	−0.191 **	0.536 **	0.591 **

BMI = body mass index; WC = waist circumference; WHR = waist-to-hip ratio; PA = physical activity; PF = physical function; HRQoL = health-related quality of life; ns = not significant; * *p* < 0.010; ** *p* < 0.001.

**Table 3 ijerph-19-13718-t003:** Results of linear regression analysis models for BMI and HRQoL.

Variable	Model 1			Model 2		
	OR	95% CI	Value	OR	95% CI	*p*-Value
BMI	−0.091	−0.630–0.114	<0.001	−0.069	−0.534–0.025	0.032
PA	0.055	0.864–1.762	0.038	0.082	0.231–2.129	0.015
PF	0.453	0.047–0.063	<0.001	0.400	0.039–0.057	<0.001

BMI: body mass index; PA: physical activity; PF: physical function; Model 1: unadjusted; Model 2: adjusted by sex, age, comorbidities.

## Data Availability

The data presented in this study are available upon request from the corresponding author.

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
