# Peer review of "The Mediating Role of Physical Activity and Physical Function in the Association between Body Mass Index and Health-Related Quality of Life: A Population-Based Study with Older Adults"

_ijerph, 2022, doi:10.3390/ijerph192113718_

Round 1

Reviewer 1 Report

According to the authors, the objective of this study was to evaluate the mediating role of PA and PF in the relationship between BMI and HRQoL, assessing this issue in an older adult population residing in the Autonomous Region of Madeira, Funchal, Portugal. Even though interesting data were presented there are another couple of factors that should be concerned.

In the Introduction section:

1) Although the authors presented important information about the context addressed in this study, some pieces of information are redundant and repetitive. Therefore, I suggest revising this section in order to remove the repetitive information and be more concise with these pieces of information.

2) Please, add the word “physical” to address the exercise on page 2 and line 94.

In the Material and Methods section:

3) Please, check the meaning of "trained researchers". (Page 3, Lines 135-136).

4) When was the study performed?

5) The authors declared that it was assessed some comorbidities, but they did not clarify if was only the occurrence of hypertension and diabetes were evaluated, or if others, such as metabolic syndrome and sarcopenia were also assessed. In addition, older adults with cancer were included?

6) What was the reference used to classify the BMI values?

7) What means SFT? (Page 4, Line 172)

In the Results section

8) Please, include in the table all the clinical characteristics of the population enrolled in the study, especially those associated with functional capacities, such as the occurrence of dynapaenia, sarcopenia, and frailty.

In the Discussion section:

9) Despite the BMI being broadly used, it is important that the authors provide a discussion concerning its use since this index cannot reflex the real situation of the individual, in terms of adiposity. In this sense, I would like to know, if there was any association between BMI and WC values in the volunteers assessed here?

10) Although the study appears to have been carried out before the COVID-19 pandemic, it would be important to add some considerations about the effects of the pandemic in the context of the study.

Author Response

Dear Reviewer, we are grateful for all the comments, and are available for future clarifications and/or corrections.

* Changes referring to the last review of the manuscript were carried out in the text using Microsoft Word's built-in track changes function.

Reviewer 1

A) Introduction

1) Although the authors presented important information about the context addressed in this study, some pieces of information are redundant and repetitive. Therefore, I suggest revising this section in order to remove the repetitive information and be more concise with these pieces of information.

2) Please, add the word “physical” to address the exercise on page 2 and line 94.

 Reply

Dear Reviewer, the word has been added.

B) Material and Methods:

3) Please, check the meaning of "trained researchers". (Page 3, Lines 135-136).

Reply

Dear Reviewer, expression has been adjusted

4) When was the study performed?

Reply

Dear Reviewer, this information was included in section 2.1.

5) The authors declared that it was assessed some comorbidities, but they did not clarify if was only the occurrence of hypertension and diabetes were evaluated, or if others, such as metabolic syndrome and sarcopenia were also assessed. In addition, older adults with cancer were included?

Reply

Dear Reviewer, we have included only the most prevalent comorbidities in the study in Table 1. Regarding cancer, we inform in the exclusion criteria that individuals with this disease did not participate in the study. Other information questioned and not evaluated by the study was detailed in the study limitations section.

6) What was the reference used to classify the BMI values?

Reply

Dear Reviewer, initially, the BMI classification was performed using the WHO cut-off points. However, after the recommendations of Reviewer 2, we adopted the values of Lipschitz (1992) that are more suitable for the older adult population. Thus, the number of participants classified as normal weight rose from 13% to 26%.

 7) What means SFT? (Page 4, Line 172)

Reply

Dear Reviewer, "SFT" is the name of the battery of functional tests used: Senior Fitness Test. This abbreviation was introduced on line 162.

C) Results

1) Please, include in the table all the clinical characteristics of the population enrolled in the study, especially those associated with functional capacities, such as the occurrence of dynapaenia, sarcopenia, and frailty.

 Reply

Dear Reviewer, this information was not collected by the study. On the other hand, they were recognized as limitations of our study. Furthermore, we suggest that future studies include these variables in their analyses.

D) Discussion

1) Despite the BMI being broadly used, it is important that the authors provide a discussion concerning its use since this index cannot reflex the real situation of the individual, in terms of adiposity. In this sense, I would like to know, if there was any association between BMI and WC values in the volunteers assessed here?

Reply

Dear Reviewer, discussion of BMI deficits to assess adiposity has been included in the text. Furthermore, following suggestions from Reviewer 2, we added Pearson's correlation analyses to verify relationships between BMI and sex/age. Parallel to the fact, we verified an association between BMI and WC. Moreover, to broaden the understanding of the nutritional status of the participants, we added the WHR-waist-to-hip ratio variable to the analyses.

2) Although the study appears to have been carried out before the COVID-19 pandemic, it would be important to add some considerations about the effects of the pandemic in the context of the study.

Reply

Dear Reviewer, as per your suggestion, we have included in the Conclusion section considerations on the relationship between obesity, older adult, and Covid-19

Reviewer 2 Report

The work deals with the interesting topic of PA and overweight and obesity in the elderly. The authors examined a very large group, but did not use it. They admit that their results confirm earlier published studies by other authors. There is no innovative element in the work, apart from the introduction of models 1 and 2. However, they have not been described clearly enough to be repeated.

The results of lines 259-267 are unclear, it is not known what is actually going on.

Why was the group not divided according to gender - if the authors noticed a relationship between the obtained results and the sex of the respondents. Both in everyday life and in sports in the elderly age, PA depends largely on the gender of the respondents.

In Table 1, the authors show that the group of people with a normative body weight constituted only 13.6%, which is doubtful if the research was carried out on the population of seniors living in a given region and not directed to the treatment of overweight or obesity. The problem of the lack of a normal distribution may result from the incorrect assumption of the classification of incorrect body mass, taking into account the BMI index. Many authors indicate that in the elderly age, health well-being is related to BMI within 27 kg / m2.

The work requires changes in the methodology in its present form is only information generally known that as BMI increases, PA decreases.

Author Response

Dear Reviewer, we are grateful for all the comments, and are available for future clarifications and/or corrections.

* Changes referring to the last review of the manuscript were carried out in the text using Microsoft Word's built-in track changes function.

Review 2

The work deals with the interesting topic of PA and overweight and obesity in the elderly. The authors examined a very large group but did not use it. They admit that their results confirm previous studies published by other authors. There is no innovative element in the work, other than the introduction of models 1 and 2. However, they were not described clearly enough to be repeated.

1) Os resultados das linhas 259-267 não são claros, não se sabe o que realmente está acontecendo.

Reply

Dear Reviewer, this passage has been rewritten. We hope to have clarified your doubts.

2) Por que o grupo não foi dividido de acordo com o gênero - se os autores perceberam uma relação entre os resultados e o sexo dos entrevistados. Tanto na vida cotidiana quanto no esporte na terceira idade, a AF depende muito do sexo dos entrevistados.

Reply

Dear Reviewer, thank you for your valuable comment. To better clarify your doubts, we have added a correlation analysis in the current version of the manuscript. Thus, in Table 2, we present the degree of association between sex and age with anthropometric variables, level of physical activity, and physical function: all results indicated small correlation coefficients. Only the HRQoL variable showed a medium correlation with sex, although it was borderline. For this reason, we chose not to analyze the main characteristics of the sample according to sex (Table 1). Thus, as we have assumed the Lipschitz BMI classification (following his suggestion), we have expanded the analysis by comparing the main variables, according to the new BMI classifications: underweight, normal weight, overweight.

3) Na Tabela 1, os autores mostram que o grupo de pessoas com peso corporal normativo compreende apenas 13,6%, o que é duvidoso se a pesquisa foi realizada na população de idosos residentes em determinada região e não direcionada ao tratamento do excesso de peso ou obesidade. O problema da falta de distribuição normal pode resultar da suposição incorreta da classificação de massa corporal incorreta, levando em consideração o índice de IMC. Muitos autores indicam que na terceira idade, o bem-estar da saúde está relacionado ao IMC dentro de 27 kg/m2.

Reply

Dear Reviewer, in relation to this observation, it really makes sense! For this reason, we assumed the BMI classification suggested by Lipschitz (1992), and redesigned the analyses. Thus, the number of older adults classified with the normal weight rose from 13% to 26%. Another change was the comparison of variables according to nutritional status (Table 1).

4) O trabalho requer mudanças na metodologia em sua forma atual é apenas uma informação de conhecimento geral que à medida que o IMC aumenta, a PA diminui.

Reply

Dear Reviewer, after the changes made, we hope to have expanded and qualified the study. We are available for future adjustments if required.

Round 2

Reviewer 2 Report

The authors took my comments into account.In this version, the work is suitable for printing and will supplement the knowledge on the role of physical activity in the aging process